# A Numerical Procedure for Multivariate Calibration Using Heteroscedastic Principal Components Regression

**Alessandra da Rocha Duailibe Monteiro [1], Thiago de Sá Feital [2,3] and José Carlos Pinto [2,*]**

[1] Departamento de Engenharia Química e Petróleo (TEQ), Universidade Federal Fluminense, Niterói, Rio de Janeiro CEP 24210-240, RJ, Brazil; alessandra_duailibe@id.uff.br or alessandrardmonteiro@gmail.com

[2] OPTIMATECH LTDA, Rio de Janeiro CEP 21941-614, RJ, Brazil; tfeital@peq.coppe.ufrj.br or thiago.feital@optimatech.solutions

[3] Programa de Engenharia Química/COPPE, Universidade Federal do Rio de Janeiro, Cidade Universitária, CP 68502, Rio de Janeiro CEP 21941-972, RJ, Brazil

[*] Correspondence: pinto@peq.coppe.ufrj.br; Tel.: +55-21-2562-8337

**Abstract:** Many methods have been developed to allow for consideration of measurement errors during multivariate data analyses. The incorporation of the error structure into the analytical framework, usually described in terms of the covariance matrix of measurement errors, can provide better model estimation and prediction. However, little effort has been made to evaluate the effects of heteroscedastic measurement uncertainties on multivariate analyses when the covariance matrix of measurement errors changes with the measurement conditions. For this reason, the present work describes a new numerical procedure for analyses of heteroscedastic systems (heteroscedastic principal component regression or H-PCR) that takes into consideration the variations of the covariance matrix of measurement fluctuations. In order to illustrate the proposed approach, near infrared (NIR) spectra of xylene and toluene mixtures were measured at different temperatures and stirring velocities and the obtained data were used to build calibration models with different multivariate techniques, including H-PCR. Modeling of available xylene–toluene NIR data revealed that H-PCR can be used successfully for calibration purposes and that the principal directions obtained with the proposed approach can be quite different from the ones calculated through standard PCR, when heteroscedasticity is disregarded explicitly.

**Keywords:** heteroscedastic principal components regression (H-PCR); measurement error; multivariate analysis; numerical procedure; near infrared spectroscopy (NIRS)





## 1. Introduction

Multivariate calibration methods constitute indispensable tools for solving many "real-world" problems [1]. Nevertheless, although it has long been recognized that measurement errors are inherent components of experimental measurements, traditional multivariate calibration methods, such as principal components analysis (PCA), principal components regression (PCR), partial least squares (PLS) and parallel factor analysis (PARAFAC), implicitly assume the occurrence of independent and identically distributed measurement Gaussian errors [2]. Alternative multivariate calibration techniques, including multiple linear regression (MLR), continuum regression (CR), projection pursuit regression (PPR), locally weighted regression (LWR) and artificial neural network modeling (ANNs), among others, implicitly assume similar measurement error conditions [1].

PCA and PCR can possibly be regarded as the most popular and most powerful chemometric tools for process monitoring and quantitative analyses [3–6]. Initially employed by statisticians to describe the variance and covariance of random variables, PCA is more commonly used in chemometrics to describe determinist relationships among variables, especially in cases where a high degree of collinearity exists or in cases of process datasets with missing values [7]. According to the PCA technique, the number of variables

of the problem can be reduced through suitable linear combinations, so that variable combinations concentrate the highest possible variance of the available data [8,9]. Then, the new set of uncorrelated variables, called principal components, spans a space of lower dimension. Reviews of PCA and some important and traditional applications can be seen elsewhere [10]. Therefore, the objective of PCA is generally twofold: to determine the best calibration model with the smallest number of variables [5].

Partial least squares (PLS) is a technique used for construction of predictive models when large numbers of measurements are available and the variables present strong collinearity [11]. The main objective of PLS is to extract latent variables from the available measurements, optimizing the information content for construction of the calibration model. In order to do that, a multidimensional direction is defined in the input X-space for maximization of the correlation with response variables in the Y-space. Therefore, PLS models can be applied when the number of measurement variables is higher than the observed measurements and when the measurements are multicollinear [11,12].

Unfortunately, the use of MLR, PCA or PLS procedures is oftentimes inadequate for calibration purposes and estimation of model parameters, as these methods rely on assumptions that impose limitations on the use of these techniques [13]. For example, these techniques assume that the analyzed variables are correlated linearly and are subject to random Gaussian fluctuations. Moreover, these techniques implicitly assume that measurement errors do not change in the experimental region and that input variables are not subject to measurement errors. Hong et al. (2018) observed that PCA performance for high-dimensional heteroscedastic data was worse than for homoscedastic data [14]. In order to overcome some of these limitations, many distinct numerical procedures have been proposed in the literature, including techniques based on maximum likelihood criteria, such as the MLPCR (maximum likelihood principal component regression) procedure [3].

It is important to emphasize that near infrared spectroscopy (NIRS) is a very robust technique for online monitoring and control of industrial processes, allowing for remote measurement of many useful process variables [1,15]. In particular, NIR-based monitoring procedures have been boosted by the continuous improvement of spectroscopic methods and fiber optics technology, which allow for the in situ and in-line acquisition of process data [15]. As NIR spectra contain significant amounts of information regarding chemical composition and phase morphology of mixtures and reacting systems, NIR spectroscopy finds applications in many different areas. Besides, as NIR-based procedures are fast, non-destructive, non-invasive, allow for direct in-line measurements and require minimum sample preparation, NIR technologies became particularly important for control, monitoring and optimization of industrial processes [16]. However, given the usual complex nature of NIR spectra, the advancement of computational resources, both in terms of hardware and software, was of fundamental importance for implementation of the calibration models that make the use of NIR possible in the industrial field [15].

Fluctuations in the experiments due to sample preparation, operation procedures variations and environment oscillations, among other factors, are quite common. However, these perturbations are not necessarily independent from each other and, when studying NIR measurements, they can strongly affect the performances of NIR spectrometers, requiring explicit consideration of the error structure in the proposed modeling approach [15]. Some real perturbations in NIR measurements can be, for example, light source fluctuations, temperature oscillations and mechanical vibration [17,18]. It is noteworthy that fluctuations of certain variables can affect the behavior of other variables, as one can easily understand when temperature and pressure are measured simultaneously in a pressurized vessel—as temperature increases, pressure is also expected to increase, meaning that temperature and pressure fluctuations cannot be independent in a closed vessel. Although light absorbance fluctuations in NIR experiments can be strongly correlated with neighboring wavelengths (as shown by Monteiro et al., 2017) [18], the importance of measurement error fluctuations (characterized in terms of the covariance matrix of the spectral responses, as functions of the measurement conditions and measured experimentally through replication) on NIR

calibration modeling has been largely overlooked in the literature. Similar phenomena can also affect many other processes and measuring conditions.

Shah and Narasimhan (2007) [19] developed an interactive algorithm for simultaneous identification of the model and the errors, using the maximum likelihood principal component analysis technique. In simple words, the method proposed the estimation of the calibration model and of the covariance matrix of measurement errors simultaneously, using the same dataset. The algorithm was developed for the situation when measurement errors in the different variables were not the same and correlated to each other. However, Santos et al. (2005) [3] showed that the simultaneous estimation of model parameters and measurement errors can eventually lead to serious numerical and statistical interpretation problems, as hard constraints must be imposed on the estimated covariances in order to assure the positive definiteness of the covariance matrix of measurement errors.

As a matter of fact, the quality and performance of multivariate calibration models depend on the effective treatment of errors. Specifically, when spectral data are considered, variances of measurement fluctuations may be different at distinct wavelengths and may be correlated to each other. For this reason, a MLPCR method was proposed by Wentzell et al. (1997) [20] in order to compensate for these effects and provide more accurate multivariate calibration models. However, the proposed approach requires either complete knowledge of the covariance matrix of measurement errors or full experimental characterization of the covariance matrix of measurement uncertainties through replication.

Bhatt et al. (2005) [16] proposed a method for development of multivariate calibration models from non-replicated measurements when errors in different absorbances are independent but can be subject to different unknown variances. The method, named iterative principal component analysis, also estimates the lower dimensional spectral subspace and all the corresponding error variances simultaneously. Wentzell et al. (2005) [21] developed a systematic approach for characterizing the covariance matrix of measurement fluctuations for particular experimental or instrumental environments. The approach was applied to different spectral systems (including UV-VIS absorption, NIR reflectance, fluorescence emission and short-wave NIR absorption) and it was noted that more detailed characterization of the error structures can bring numerous benefits to the analysis, including the enhanced performance of calibration models. Hong et al. (2020) [22] developed a probabilistic PCA model that incorporates the heteroscedastic noise data and derives an expectation maximization algorithm to compute the factor estimate. Homoscedastic PCA was applied to initialize the algorithm.

It is important to highlight, though, that previous studies have systematically neglected the fact that the full covariance matrix of measurement fluctuations (and not only the variances, or the elements of the main diagonal of the covariance matrix [14,22]) can change with the measurement condition (depending on other variables besides the spectral wavelength, in the case of NIR calibrations). In other words, the analyzed system may be heteroscedastic and the covariance matrix of measurement responses (or the spectral output, measured in terms of absorbance, transmittance, reflectance or other suitable response variable, in the case of NIR calibrations) can depend on the wavelength, but also on concentrations and temperatures, among other variables. Particularly, it must be highlighted that the heteroscedastic nature of the measurement fluctuations has been used to provide information about the kinetic behavior of chemical reaction systems as a function of the reaction conditions [23,24], affecting the estimation of kinetic parameters.

To the best of our knowledge, procedures that consider the variations of the covariance matrix of measurement fluctuations along the experimental grid have not been used for calibration purposes (and therefore for NIR-based model building); consequently, the effects of changing covariance matrixes of measurement fluctuations on model calibration and model performance have not been analyzed yet (including NIR calibration problems). Despite that, it is important to recognize that previous studies have attempted to reduce the sensitivity of calibration models to unknown measurement perturbations using different pretreatment techniques [14,25–34], although not based on the detailed statistical charac-

terization of the measurement fluctuations, as described in terms of variance spectra and covariance matrixes of measurement fluctuations calculated with replicates under distinct experimental conditions. In addition, cluster analyses and other grouping procedures have been used to account for the existence of data subsets that may be subject to distinct statistical behavior [35], which is not equivalent to taking into consideration the characteristic covariance matrix of measurement fluctuations for each particular experimental point used for model building and calibration.

Monteiro et al. (2017) [18] presented a statistical study regarding some simple NIR experiments, characterizing the importance of variances and covariances of measurement errors for quantitative NIR analyses. Calibration models were built with help of different multivariate techniques (MLR, PCR and PLS) [20]. The authors showed that the existence of varying measurement fluctuations and measurement error correlations along the experimental grid can significantly affect the model calibration step and the final model performance.

As a matter of fact, the effects of varying measurement fluctuations and measurement error correlations on model building procedures and model prediction errors have been systematically neglected in calibration problems, including NIR applications. Little effort has been made to evaluate the covariances of measurement uncertainties through experiments in the literature and to introduce such covariances in the modeling step, particularly when they are subject to changes in the experimental grid. Numerical procedures have been more frequently proposed to estimate covariance matrixes of measurement errors and to allow for the simultaneous estimation of covariances of measurement fluctuations and calibration model parameters when sufficiently large sets of industrial data are available [22,35,36]. However, these procedures are not based on the independent characterization of the covariance matrix of experimental fluctuations as a function of the experimental conditions and have not been used yet for NIR model calibrations and characterization of measurement error correlations in NIR experiments.

Particularly, non-conventional calibration procedures have usually proposed the estimation of the covariance matrix of measurement fluctuations and of the model parameters simultaneously [37], although these strategies do not seem appropriate when the experimental variances and covariances change at each distinct experimental points. Alternative strategies may consider the systematic experimental investigation of the measurement errors [2] and the use of maximum likelihood estimation procedures for model building [1], which was the strategy pursued here. For this reason, in the present work, a heteroscedastic principal component regression (H-PCR) method was developed to deal with calibration problems and is compared to other traditional calibration techniques (CLS, PCR and PLS) in a simple calibration problem, related to use of NIR spectra for monitoring of concentrations of xylene–toluene mixtures [18]. Modeling of available xylene–toluene NIR data revealed that H-PCR can be used successfully for calibration purposes and that the principal directions obtained with the proposed approach can be quite different from the ones calculated through standard PCR, when heteroscedasticity is disregarded explicitly.

## 2. Methodology

### 2.1. Theoretical Framework

The standard multivariate calibration process consists of building a model to correlate $n$ spectra (the number of experimental points), comprising $m$ distinct wavelengths (the size of the input vector $\mathbf{x}_m$), to $n$ response values (the response variable y) obtained through references or independent methods (usually concentrations) [20]. The calibration procedure begins with the construction of the input data matrix, $\mathbf{X}_{mxn}$, which contains the collected NIR spectra, and the data vector $\mathbf{y}_n$, which contains the output responses. The calibration model must provide a vector $\mathbf{y}^c_n$, containing values calculated with $\mathbf{X}_{mxn}$ and that is expected to be sufficiently close to vector $\mathbf{y}_n$. The most common multivariate calibration techniques that are used to build NIR calibration models are the classical least

squares (CLS), multiple linear regression (MLR), partial least squares (PLS) and principal component regression (PCR), as briefly described below.

The standard CLS technique proposes the representation of the original multivariable problem as a set of univariate problems for each of the considered analytes, in the form:

$$\mathbf{y} = a + b\,\mathbf{x}_1 + \mathbf{e} \tag{1}$$

where $\mathbf{y}$ is the vector of model responses (usually concentrations) and $\mathbf{x}_1$ is a vector of inputs (usually the spectral responses at a defined wavelength). $a$ and $b$ are model parameters, respectively, linear coefficient or axis interception and angular coefficient or slope, which must be determined through model fitting, using the available experimental data. $\mathbf{e}$ is a vector of residuals, containing the experimental measurement fluctuations and the possible model inadequacy. If the calibration model is efficient, $\mathbf{e}$ should have zero mean and variance similar to the variance of $\mathbf{y}$ measurements, normally assumed to be constant in the whole experimental grid (homoscedasticity). Therefore, the CLS procedure requires the definition of the concentrations of all the spectroscopically active species and of the respective wavelengths that must be used for purposes of model building, which is not always possible in practical situations [38]. However, it must be acknowledged that calibration of dilute solutions of non-interacting chemical species may rely on spectral responses of the pure materials only. The CLS procedure described here can certainly be used to represent other similar calibration problems.

The MLR technique is the simplest multivariate calibration method, which does not require the decomposition of the original problem into smaller univariate calibration problems. The method assumes that the response variable depends on multiple input variables, in the form:

$$\mathbf{y} = a + b_1\,\mathbf{x}_1 + b_2\,\mathbf{x}_2 + ...+ b_m\,\mathbf{x}_m, + \mathbf{e} \tag{2}$$

$$(\mathbf{y} - \boldsymbol{a}) = \begin{bmatrix} y_1 - a \\ \vdots \\ y_n - a \end{bmatrix} = \begin{bmatrix} x_{11} & \cdots & x_{1m} \\ \vdots & \ddots & \vdots \\ x_{n1} & \cdots & x_{nm} \end{bmatrix} \cdot \begin{bmatrix} b_1 \\ \vdots \\ b_m \end{bmatrix} = \mathbf{X}^T.\,\boldsymbol{\alpha} \tag{3}$$

where $a$ and $b_i$ are model parameters (contained in $\boldsymbol{\alpha}$) that must be determined through model fitting, using the available experimental data. When the multiple linear regression procedure is applied, the user must define a priori the set of wavelengths that must be used for calibration purposes. For this reason, the MLR technique is usually applied when the most significant fluctuations of the spectra are concentrated in narrow spectral regions [39]. Specifically, the MLR method can be very sensitive to collinearity of spectral responses, which makes parameter estimation more difficult and model performance poorer [39]. As assumed in the previous case, if the calibration model is efficient, $\mathbf{e}$ should have zero mean and variance similar to the variance of $\mathbf{y}$ measurements, normally assumed to be constant in the whole experimental grid (homoscedasticity).

Assuming that the model is perfect, that experimental measurements are subject to homoscedastic normal fluctuations and that input variables are free of error (which is not supported by independent experimental statistical analyses, as shown by Monteiro et al., 2017) [18], the model parameters must be estimated in the form [20]:

$$\min \mathbf{F} = (\mathbf{y}^e - \mathbf{y}^c)^T.\mathbf{V}_y^{-1}.\,(\mathbf{y}^e - \mathbf{y}^c) = \left(\mathbf{y}^e - \mathbf{X}^T.\boldsymbol{\alpha}\right)^T.\mathbf{V}_y^{-1}.\left(\mathbf{y}^e - \mathbf{X}^T.\boldsymbol{\alpha}\right) \tag{4}$$

$$\boldsymbol{\alpha} = \left(\mathbf{X}.\mathbf{V}_y^{-1}.\mathbf{X}^T\right)^{-1}.\left(\mathbf{X}.\mathbf{V}_y^{-1}.\mathbf{y}^e\right) \tag{5}$$

If the variances are the same throughout the experimental grid and the measurement correlations are null, then:

$$\boldsymbol{\alpha} = \left(\mathbf{X}.\mathbf{X}^T\right)^{-1}.(\mathbf{X}.\mathbf{y}^e) \tag{6}$$

which is the usual form of the MLR calibration, which can be easily extended to describe other similar problems. According to Equations (6) and (7), it can be noted that the MLR technique requires the inversion of matrix $\mathbf{X}.\mathbf{X}^T$, which explains the sensitivity to collinearity and measurement correlations. For this reason, PCR and PLS are more efficient methods to deal with large datasets [40].

The PLS technique assumes that a smaller set of latent variables affect the available experimental responses, allowing for removal of undesirable collinearity effects and for optimization of the information content of the model [11]. Initially, the normalization of the measured variables is usually performed in order to remove the units of the measured data and make the ranges of measurement variations more uniform. Then, it is acknowledged that one direction $\mathbf{p}_1$ concentrates the correlation between $\mathbf{y}$ and $\mathbf{X}$. Therefore, the estimation problem becomes:

$$\min \mathbf{F} = \left(\mathbf{y}^e - \mathbf{X}^T.\mathbf{p}_1.\alpha_1\right)^T.\mathbf{V}_y^{-1}.\left(\mathbf{y}^e - \mathbf{X}^T.\mathbf{p}_1.\alpha_1\right) \tag{7}$$

$$\alpha_1 = \left(\mathbf{p}_1^T.\mathbf{X}.\mathbf{V}_y^{-1}.\mathbf{X}^T.\mathbf{p}_1\right)^{-1}.\left(\mathbf{p}_1^T.\mathbf{X}.\mathbf{V}_y^{-1}.\mathbf{y}^e\right) \tag{8}$$

Inserting Equation (8) into Equation (7), the fundamental PLS problem can be obtained, which is the manipulation of vector $\mathbf{p}_1$ for minimization of the objective function F, which can be performed with the help of numerical procedures [11]. The effects of direction $\mathbf{p}_1$ on the calibration problem can then be removed in the form:

$$\mathbf{X}^{(1)} = (\mathbf{I} - \mathbf{p_1}^T.\mathbf{I}).\mathbf{X} \tag{9}$$

$$\mathbf{y}^{(1)} = \mathbf{y}^e - \alpha_1.\mathbf{X}^T.\mathbf{p}_1 \tag{10}$$

resulting in residuals $\mathbf{X}^{(1)}$ and $\mathbf{y}^{(1)}$, which can be used for determination of $\alpha_2$ and $\mathbf{p}_2$ in a similar manner. The procedure can be repeated iteratively in order to maximize the correlation between $\mathbf{X}$ and $\mathbf{y}$ [11,40,41]. One must observe that Equations (7)–(10) implicitly assume the validity of the same hypotheses described earlier, regarding the fact that $\mathbf{X}$ is free of error in the whole experimental range.

The PCR technique can be regarded as a simplification of the PLS technique, obtained by assuming that $\mathbf{p}_i$ vectors are the directions that concentrate the largest possible variance of the available data [10,42,43], usually computed in the form:

$$\mathbf{X}.\mathbf{X}^T = \mathbf{D}.\mathbf{\Lambda}.\mathbf{D}^T \tag{11}$$

where $\mathbf{\Lambda}$ is the diagonal matrix that contains the eigenvalues $\lambda_1, \lambda_2,..,\lambda_m$ of $\mathbf{X}.\mathbf{X}^T$ and $\mathbf{D}$ is the matrix that contains the eigenvectors $\mathbf{v}_1, \mathbf{v}_2,....,\mathbf{v}_m$ of $\mathbf{X}.\mathbf{X}^T$ [44]. Then, the $\mathbf{p}_i$ vectors are equal to the eigenvectors $\mathbf{v}_i$, associated with the largest eigenvalues $\lambda_1 > \lambda_2 >... > \lambda_M >... > \lambda_m$, where M defines the size of the proposed model and the number of latent variables used for calibration purposes. According to the PCR method, the directions used to construct the model concentrate a fraction ($\varphi$) of the total experimental variability, defined in the form:

$$\varphi = \frac{\sum_{i=1}^{M} \lambda_i}{\sum_{j=1}^{m} \lambda_j} \tag{12}$$

One must observe once more that the PCR technique implicitly assumes the validity of the same hypotheses described earlier, regarding the fact that $\mathbf{X}$ is free of error in the whole experimental range. More detailed discussions about PCR and PLS techniques can be found elsewhere [45,46].

Maximum likelihood principal component regression (MLPCR) is a decomposition method that resembles the conventional PCR, but that takes into account measurement uncertainty during the decomposition process, placing less emphasis on measurements with large variances [3,47]. It is important to recognize that MLPCR is not just a variation of PCR, but a more general numerical procedure for multivariate modeling [47].

The first step of the MLPCR technique consists of determining the direction of maximum variability of the independent variables. In this context, Equation (4), which defines the objective function for multivariate calibration procedures, can be presented in the form:

$$\min \mathbf{F} = \sum_{k=1}^{M} \sum_{i=1}^{n} \left( \mathbf{X}_i - \left( \mathbf{X}_i^T . \mathbf{p}_k \right) \mathbf{p}_k \right)^T . \mathbf{V}_{\mathbf{X}_i}^{-1} . \left( \mathbf{X}_i - \left( \mathbf{X}_i^T . \mathbf{p}_k \right) \mathbf{p}_k \right) \tag{13}$$

which depends on the covariance matrixes of measurement fluctuations of $\mathbf{X}_i$. Matrix $\mathbf{V_{xi}}$ should contain the real experimental fluctuations of the spectral measurements in order for use of Equation (13) to make sense. The problem, though, is that matrix $\mathbf{V_{xi}}$ is normally singular (and, therefore, not invertible) in real calibration problems, as the wavelength range is normally large (*m* is a large number, in the range of hundreds) and the number of replicates (*NR*) used for independent characterization of experimental errors is much smaller than *m* (*NR* is almost always smaller than 10). A similar limitation can be observed in the vast majority of large calibration datasets. Therefore, Equation (13) cannot be used in the proposed form. Moreover, $\mathbf{V_{xi}}$ can depend on the measurement conditions and cannot be efficiently estimated from calibration residuals [15].

### 2.2. The Proposed Heteroscedastic Technique

The model calibration procedure based on the proposed heteroscedastic PCR technique consists, essentially, of inserting the evaluated covariance matrixes of measurement fluctuations, characterized at each measurement condition, directly into Equation (13). In order to perform the minimization task described in Equation (13), a stochastic algorithm has been devised, associating a set of plausible measurements to each experimental condition, in accordance with specific generated probability values [46]. Random number generation procedures are used to generate the problem solution candidates, as described below.

As presented in Equation (13) and described previously, the covariance matrixes of measurement fluctuations identified experimentally are expected to be singular. Therefore, it is convenient to assume that the matrixes $\mathbf{V_{xi}}$ can be decomposed in the form $\mathbf{V_{xi}} = \mathbf{D_i}.\mathbf{\Lambda_i}.\mathbf{D_i}^T$, where $\mathbf{\Lambda_i}$ is the diagonal matrix that contains the eigenvalues and $\mathbf{D_i}$ is the matrix that contains the eigenvectors of $\mathbf{V_{xi}}$. Matrixes $\mathbf{V_{xi}}$, i= 1 ... *n*, are known, as calculated with number of replicates $NR < m$, assuming that independent statistical characterization of measured data has been performed and that the covariance matrixes of measurement fluctuations are available, as shown by Monteiro et al. (2017) [18]. Equation (13), however, requires calculation of $\mathbf{V_{xi}}^{-1}$, although matrixes $\mathbf{V_{xi}}$, i= 1 ... *n*, are often noninvertible. Therefore, it is initially necessary to propose suitable pseudo-inverses for matrixes $\mathbf{V_{xi}}$, i= 1 ... *n*, which are calculated here in the form [48]:

$$\mathbf{V}_{xi}^{-1} = \mathbf{D_i}.\mathbf{\Lambda_i}^{-1}.\mathbf{D_i}^T \tag{14}$$

where $\mathbf{\Lambda}_i$ is a diagonal matrix that contains the positive eigenvalues of $\mathbf{V}_{xi}$ in the main diagonal and $\mathbf{D}_i$ contains the respective eigenvectors in the columns. Equation (14) then requires calculation of eigenvalues and eigenvectors of the error covariance matrix when it is decomposed. Specifically, in the present study matrix, $\mathbf{V_{xi}}$ has been characterized experimentally using NIR spectra of solutions containing varying amounts of xylene and toluene at distinct measurement conditions, confirming the small number of nonsingular eigenvalues of $\mathbf{V_{xi}}$ and, consequently, the small number of variability sources in the analyzed problem.

When the pseudo-inverse matrixes defined in Equation (14) are inserted into the objective function equation (Equation (13)), it becomes possible to estimate the directions $\mathbf{p}_k$ that concentrate the maximum variability of the available data, as weighed by the measured covariance matrixes of measurement fluctuations. In order to overcome numerical issues related to the high dimensionality and non-invertible nature of $\mathbf{V_{xi}}$ during analytical computations, an iterative stochastic procedure is proposed for calculation of the main

directions that constitute the columns of matrix **P**. The proposed step-by-step calculation is summarized in Equations (15)–(24). First, it is assumed that the initial candidates for the principal directions can be described in the form:

$$p_{k,i}^{(j,t)} = p_{k,ref}^{(j,t)} + a_{k,i}^{(t)} * \sin\left\{2\pi\left[\frac{f_{k,i}^{(t)} \cdot j}{m} + b_{k,i}^{(t)}\right]\right\} \tag{15}$$

where $p_{ki}^{(j,t)}$ is the *j*th component of the *i*th candidate for the principal direction $\mathbf{p}_k$, $k = 1 \dots M$, $i = 1 \dots NC$, at iteration *t*. *M* (the number of latent variables) and *NC* (the number of candidates, assumed to be equal to 2000, unless stated otherwise) are numerical parameters that must be provided by the user. $p_{k,\,refi}$ is a reference direction (assumed to be equal to 1 at iteration 1), needed for proper normalization of the calculated values; *j* is a counter, ranging from one to *m*, for the dimension of the space of independent measurements **X**. $a_{min}^{(t)} < a_{k,i}^{(t)} < a_{max}^{(t)}$, $f_{min}^{(t)} < f_{k,i}^{(t)} < f_{max}^{(t)}$ and $b_{min}^{(t)} < b_{k,i}^{(t)} < b_{max}^{(t)}$ are random numbers that follow the uniform distribution and define, respectively, the amplitude, the frequency and the lag of a characteristic sinusoidal signal. Equation (15) assumes that the principal directions can be represented as smooth functions of the wavelength, as in usual Fourier decompositions [49]. Equation (15) suggests that the principal direction can be represented by a sum of random signals with different amplitudes and frequencies. Next, the maximum likelihood objective function presented in Equation (13) can be calculated for each candidate of principal direction, as presented in Equation (16):

$$F_{k,i}^{(t)} = \sum_{i=1}^{n} \left(\mathbf{X}_i - \sum_{l=1}^{k}\left[\left(\mathbf{X}_i^{\mathrm{T}}\cdot\mathbf{p}_{l,i}^{(t)}\right)\mathbf{p}_{l,i}^{(t)}\right]\right)^{\mathrm{T}} \cdot \mathbf{V}_{\mathbf{X}_i}^{-1}\cdot\left(\mathbf{X}_i - \sum_{l=1}^{k}\left[\left(\mathbf{X}_i^{\mathrm{T}}\cdot\mathbf{p}_{l,i}^{(t)}\right)\mathbf{p}_{l,i}^{(t)}\right]\right) \tag{16}$$

so that the best set of candidates for principal directions, $\mathbf{P}_{opt}^{(t)}$, can be selected as the set that leads to the lowest values of $F_{k,i}^{(t)}$ at iteration *t*, $i = 1 \dots NC$, $k = 1 \dots M$, $F_{k,opt}^{(t)}$. The procedure can then be repeated iteratively, by imposing:

$$\mathbf{p}_{k,ref}^{(t+1)} = \mathbf{p}_{k,opt}^{(t)} \tag{17}$$

$$a_{min}^{(t+1)} = \left(\frac{a_{min}^{(t)} + a_{max}^{(t)}}{2}\right) - \mu\left(\frac{a_{max}^{(t)} - a_{min}^{(t)}}{2}\right) \tag{18}$$

$$a_{max}^{(t+1)} = \left(\frac{a_{min}^{(t)} + a_{max}^{(t)}}{2}\right) + \mu\left(\frac{a_{max}^{(t)} - a_{min}^{(t)}}{2}\right) \tag{19}$$

$$f_{min}^{(t+1)} = \left(\frac{f_{min}^{(t)} + f_{max}^{(t)}}{2}\right) - \mu\left(\frac{f_{max}^{(t)} - f_{min}^{(t)}}{2}\right) \tag{20}$$

$$f_{max}^{(t+1)} = \left(\frac{f_{min}^{(t)} + f_{max}^{(t)}}{2}\right) + \mu\left(\frac{f_{max}^{(t)} - f_{min}^{(t)}}{2}\right) \tag{21}$$

$$b_{min}^{(t+1)} = \left(\frac{b_{min}^{(t)} + b_{max}^{(t)}}{2}\right) - \mu\left(\frac{b_{max}^{(t)} - b_{min}^{(t)}}{2}\right) \tag{22}$$

$$b_{max}^{(t+1)} = \left(\frac{b_{min}^{(t)} + b_{max}^{(t)}}{2}\right) + \mu\left(\frac{b_{max}^{(t)} - b_{min}^{(t)}}{2}\right) \tag{23}$$

where μ is a contracting factor that controls the convergence and must be provided by the user (made equal to 0.9, unless stated otherwise). The procedure must be halted when numerical convergence is detected:

$$\frac{\left| F_{k,opt}^{(t+1)} - F_{k,opt}^{(t)} \right|}{F_{k,opt}^{(t)}} < \delta \tag{24}$$

where $\delta$ is a numerical parameter that must be provided by the user in order to guarantee the accuracy of the final response. For practical purposes, $\delta$ was made equal to $1.0 \times 10^{-4}$ in all simulations, unless stated otherwise. As observed experimentally, the use of larger $\delta$ values can lead to inaccurate and oscillatory responses, while the use of smaller $\delta$ values can cause a significant increase in computational time without any significant improvement in the obtained numerical results.

Finally, after calculation of the principal directions, the calibration model parameters can be computed as illustrated in Equations (7)–(10), keeping the principal directions constant. The calibration modeling can be repeated for an increasing number of latent variables for determination of the best H-PCR model. The optimum number of latent variables can be determined with help of cross-validation strategies, as usually performed in the literature [20].

Alternatively, Equations (15)–(24) can be used to generate the heteroscedastic PLS algorithm. As shown in the next sections, the prediction variances of outputs are heteroscedastic because of the heteroscedasticity of inputs and estimation of model parameters [46]. Consequently, if Equation (16) is replaced by Equations (7)–(10) to define the objective function and the principal components, and if $\mathbf{V_y}$ in Equations (7) and (8) is assumed to be the covariance matrix of model outputs, then the PLS technique becomes heteroscedastic and respects the natural variability of the analyzed system.

### 2.3. Experimental

The experimental procedure was divided into two parts. In the first part, the statistical behavior of NIR data was characterized as functions of the measurement conditions, through manipulation of stirring velocities, temperatures and concentrations. These variables were manipulated because they are frequently disturbed in real reaction environments. In the second part, model calibrations were performed using CLS, PCR, H-PCR and PLS techniques to predict concentrations as functions of measured NIR spectra, as usually performed at the plant site to monitor the course of chemical reactions. These techniques were applied in a simple problem, which consisted of analyzing mixtures of xylene and toluene with help of NIR spectroscopy at different concentrations, temperatures and stirring velocities, as described in Table 1.

**Table 1.** Experiments performed for model calibration.

| Xylene Concentration [v/v%] | Toluene Concentration [v/v%] | Temperature [°C] | Stirring Speed [rpm] | | |
|---|---|---|---|---|---|
| | | 30 | 250 | 350 | 450 |
| 0 | 100 | 60 | 250 | 350 | 450 |
| | | 90 | 250 | 350 | 450 |
| | | 30 | 250 | 350 | 450 |
| 10 | 90 | 60 | 250 | 350 | 450 |
| | | 90 | 250 | 350 | 450 |
| | | 30 | 250 | 350 | 450 |
| 20 | 80 | 60 | 250 | 350 | 450 |
| | | 90 | 250 | 350 | 450 |

**Table 1.** *Cont.*

| Xylene Concentration [v/v%] | Toluene Concentration [v/v%] | Temperature [°C] | Stirring Speed [rpm] | | |
|---|---|---|---|---|---|
| 30 | 70 | 30 | 250 | 350 | 450 |
| | | 60 | 250 | 350 | 450 |
| | | 90 | 250 | 350 | 450 |
| 40 | 60 | 30 | 250 | 350 | 450 |
| | | 60 | 250 | 350 | 450 |
| | | 90 | 250 | 350 | 450 |
| 50 | 50 | 30 | 250 | 350 | 450 |
| | | 60 | 250 | 350 | 450 |
| | | 90 | 250 | 350 | 450 |
| 60 | 40 | 30 | 250 | 350 | 450 |
| | | 60 | 250 | 350 | 450 |
| | | 90 | 250 | 350 | 450 |
| 70 | 30 | 30 | 250 | 350 | 450 |
| | | 60 | 250 | 350 | 450 |
| | | 90 | 250 | 350 | 450 |
| 80 | 20 | 30 | 250 | 350 | 450 |
| | | 60 | 250 | 350 | 450 |
| | | 90 | 250 | 350 | 450 |
| 90 | 10 | 30 | 250 | 350 | 450 |
| | | 60 | 250 | 350 | 450 |
| | | 90 | 250 | 350 | 450 |
| 100 | 0 | 30 | 250 | 350 | 450 |
| | | 60 | 250 | 350 | 450 |
| | | 90 | 250 | 350 | 450 |

Xylene and toluene were used as models because they are completely miscible and present very similar NIR spectra [18], which makes the calibration process more difficult and interesting for analyses of numerical performances. Detailed descriptions of the experimental setup and experimental methods have been presented by Monteiro et al. (2017) [18]. Despite that, it is important to say that standard factorial designs were used to build the experimental grid and that spectral data were collected at random. This was possible in the present study because the performed experiments were relatively simple. It is also important to emphasize that $NR = 10$ replicates were obtained at each distinct experimental condition and used to perform the statistical analyses, so that $n = 990$ independent NIR spectra were used for numerical analyses (11 compositions, 3 temperatures, 3 stirring speeds, 10 replicates).

NIR spectra were measured in regular intervals of 2 min, using an in situ spectrophotometer NIR-6500 (NIRSystems, Inc., Silver Spring, MD, USA), working in the transflectance mode in the spectral region of 400–2500 nm. The spectra were collected using a stainless steel transflectance probe with constant pathlength of 34 mm and diameter of 19 mm, connected to the instrument through a fiber optics cable of 3 m. The fiber optics cable comprised three bundles of fibers. The light bulb contained a filament of tungsten and the light detector was based on the standard PbS technology. Data acquisition was performed with NIR Spectral Analysis Software version 3.30, a software provided by the manufacturer of the NIR spectrometer (Vision(R)). Spectra were recorded as averages

of 32 readings with precision of 0.1 nm, according to spectrophotometer specification (NIRSystem Process Analytics Manual version 1.0 NIRSystems Inc., Silver Spring, MD, USA). Spectral bandwidth was set to $10 \pm 1$ nm and the dynamic range was equal to 2–3 AU.

Calibration models were built to provide the xylene content of the analyzed organic solution as functions of obtained NIR spectra. As usual in this field, first and second derivative spectra were calculated to magnify the differences between toluene and xylene measurements and to facilitate the calibration process, since this procedure can remove base line variations, remove measurement noise and discriminate overlapping bands [18]. Derivatives were computed with second-degree interpolating polynomials and five neighboring datapoints, simultaneously providing a smoothing effect.

It is important to observe that dataset splitting techniques are normally used to provide more robust parameter estimates and model performances. In the case of the present work, all analyzed calibration procedures allowed for reliable global estimations, concentrating at least 90% of the variability of the data, indicating that robust model performances were indeed obtained, as shown in the following sections. This is due to the large number of replicates in the dataset ($NR = 10$), which was uncommon in most previous experimental works. Consequently, random splitting of the datasets leads to smaller but statistically similar datasets, which is an additional benefit of the proposed analyses. Despite that, we must emphasize that dataset splitting was indeed used for purposes of model building, as 10% of the dataset was saved for cross-validation and used for selection of the proposed calibration models and characterization of model performances.

All calculations performed in the present work were carried out in Fortran in a microcomputer (Intel(R) Core™ i7-7500 CPU 2,7 GHz, 16 Gb of RAM and 1 TB of HD, 64 bits) equipped with a data acquisition board (analog input data acquisition board, model IPC-DAS, PCI-1002H High Gain 16-Ch, screw-terminal DB-1825), used for data acquisition and temperature control.

## 3. Results and Discussion

### 3.1. Preliminary Geometrical Interpretation of H-PCR

For illustrative purposes and better understanding of the proposed H-PCR scheme, it is convenient to analyze the behavior of simple calibration problems defined in low-dimensional input spaces. For this reason, it is assumed first that Equation (16) can be rewritten for one-dimensional problems in the form:

$$F_{1,l}^{(t)} = \sum_{i=1}^{n} \frac{\left( x_i - \alpha_{1,l}^{(t)2} \, x_i \right)^2}{\sigma_i^2} \tag{25}$$

that leads to the obvious optimal solution

$$\alpha_{1,opt}^{(t)} = \pm 1 \tag{26}$$

which constitutes the trivial solution for the eigenvectors embedded in one-dimensional spaces, showing the apparent consistency of the proposed approach. Similarly, Equation (16) can be rewritten for two-dimensional problems in the form:

$$
F_{1,l}^{(t)} = \sum_{i=1}^{n} \left( \left[ \begin{array}{c} x_1 \\ x_2 \end{array} \right]_i - \left( x_{1,i} \, p_{1,l}^{(1,t)} + x_{2,i} \, p_{1,l}^{(2,t)} \right) \left[ \begin{array}{c} p_{1,l}^{(1,t)} \\ p_{1,l}^{(2,t)} \end{array} \right] \right)^T \cdot \left[ \begin{array}{cc} \sigma_{1,i}^2 & \rho_i \, \sigma_{1,i}\sigma_{2,i} \\ \rho_i \, \sigma_{1,i}\sigma_{2,i} & \sigma_{2,i}^2 \end{array} \right]^{-1} \cdot \left( \left[ \begin{array}{c} x_1 \\ x_2 \end{array} \right]_i \right.
$$
$$
\left. - \left( x_{1,i} \, p_{1,l}^{(1,t)} + x_{2,i} \, p_{1,l}^{(2,t)} \right) \left[ \begin{array}{c} p_{1,l}^{(1,t)} \\ p_{1,l}^{(2,t)} \end{array} \right] \right) \tag{27}
$$

Assuming, for illustrative purposes, that two populations of datapoints are available, with sizes $n_1$ and $n_2$, possibly obtained at two distinct operation conditions 1 and 2, then Equation (27) can be rewritten in the form:

$$
\begin{aligned}
F_{1,l}^{(t)} = & \sum_{i=1}^{n1} \left( \left( \begin{bmatrix} x_1 \\ x_2 \end{bmatrix}_i - \left( x_{1,i}\, p_{1,l}^{(1,t)} + x_{2,i}\, p_{1,l}^{(2,t)} \right) \begin{bmatrix} p_{1,l}^{(1,t)} \\ p_{1,l}^{(2,t)} \end{bmatrix} \right)^{\mathrm{T}} \cdot \begin{bmatrix} \sigma_{1,1}^2 & \rho_1\,\sigma_{1,1}\sigma_{1,2} \\ \rho_1\,\sigma_{1,1}\sigma_{1,2} & \sigma_{1,2}^2 \end{bmatrix}^{-1} \cdot \left( \begin{bmatrix} x_1 \\ x_2 \end{bmatrix}_i \right. \right. \\
& \left. \left. - \left( x_{1,i}\, p_{1,l}^{(1,t)} + x_{2,i}\, p_{1,l}^{(2,t)} \right) \begin{bmatrix} p_{1,l}^{(1,t)} \\ p_{1,l}^{(2,t)} \end{bmatrix} \right) \right) \\
& + \sum_{i=1}^{n2} \left( \left( \begin{bmatrix} x_1 \\ x_2 \end{bmatrix}_i - \left( x_{1,i}\, p_{1,l}^{(1,t)} + x_{2,i}\, p_{1,l}^{(2,t)} \right) \begin{bmatrix} p_{1,l}^{(1,t)} \\ p_{1,l}^{(2,t)} \end{bmatrix} \right)^{\mathrm{T}} \cdot \begin{bmatrix} \sigma_{2,1}^2 & \rho_2\,\sigma_{2,1}\sigma_{2,2} \\ \rho_2\,\sigma_{2,1}\sigma_{2,2} & \sigma_{2,2}^2 \end{bmatrix}^{-1} \cdot \left( \begin{bmatrix} x_1 \\ x_2 \end{bmatrix}_i \right. \right. \\
& \left. \left. - \left( x_{1,i}\, p_{1,l}^{(1,t)} + x_{2,i}\, p_{1,l}^{(2,t)} \right) \begin{bmatrix} p_{1,l}^{(1,t)} \\ p_{1,l}^{(2,t)} \end{bmatrix} \right) \right)
\end{aligned}
\tag{28}
$$

In order to highlight the main characteristics of the H-PCR procedure, it was assumed that sets 1 and 2 contained $n_1 = n_2 = 100$ and that the experimental fluctuations could be described by the following covariance matrices of measurement fluctuations:

$$
\begin{aligned}
V_x^{(1)} &= \begin{bmatrix} \sigma_{1,1}^2 & \rho_1\,\sigma_{1,1}\sigma_{1,2} \\ \rho_1\,\sigma_{1,1}\sigma_{1,2} & \sigma_{1,2}^2 \end{bmatrix} = \begin{bmatrix} 1 & 0.09 \\ 0.09 & 0.01 \end{bmatrix} \\
V_x^{(2)} &= \begin{bmatrix} \sigma_{2,1}^2 & \rho_2\,\sigma_{2,1}\sigma_{2,2} \\ \rho_2\,\sigma_{2,1}\sigma_{2,2} & \sigma_{2,2}^2 \end{bmatrix} = \begin{bmatrix} 1 & 0.0 \\ 0.0 & 0.81 \end{bmatrix}
\end{aligned}
\tag{29}
$$

Pseudo-random numbers were then generated, assuming the normal distribution with null mean value, as shown in Figure 1a. One can clearly see the existence of two distinct populations of datapoints (presented in different colors) in Figure 1a, with different error contents and different degrees of collinearity.

Figure 1b shows the most influential principal directions of the full set of available datapoints, when the standard PCA procedure and the proposed H-PCR scheme were used. One can clearly see that the H-PCR procedure weighed the information contents of the distinct data populations and calculated a principal direction that was strongly influenced by the information provided by the most precise measurements. On the other hand, the standard PCA analysis disregarded the information contents of the available data and provided a principal direction that described the overall variability of the available measurements, assuming implicitly that $V_x^{(1)} = V_x^{(2)} = I$. Nevertheless, as discussed previously, many experimental studies have shown that this assumption is usually incorrect and can lead to serious misinterpretation of the available experimental data [18,23,24].

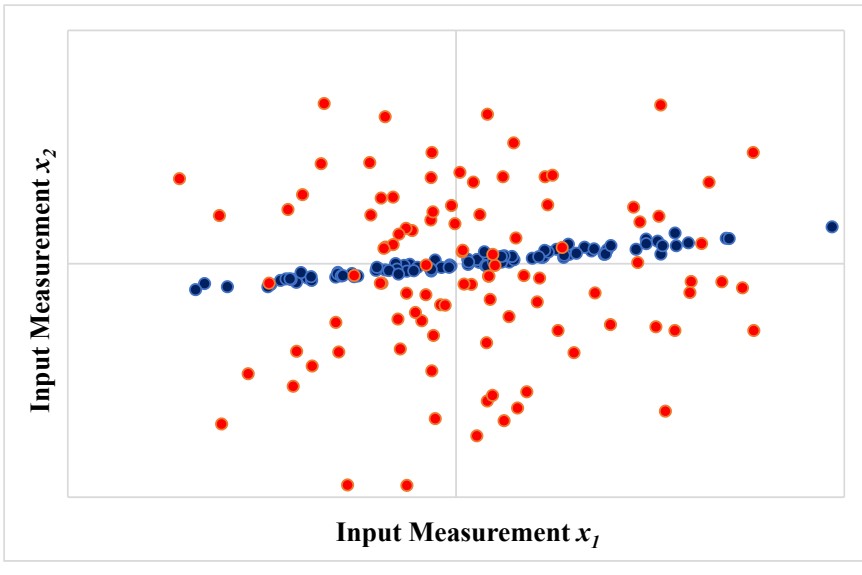

(**a**)

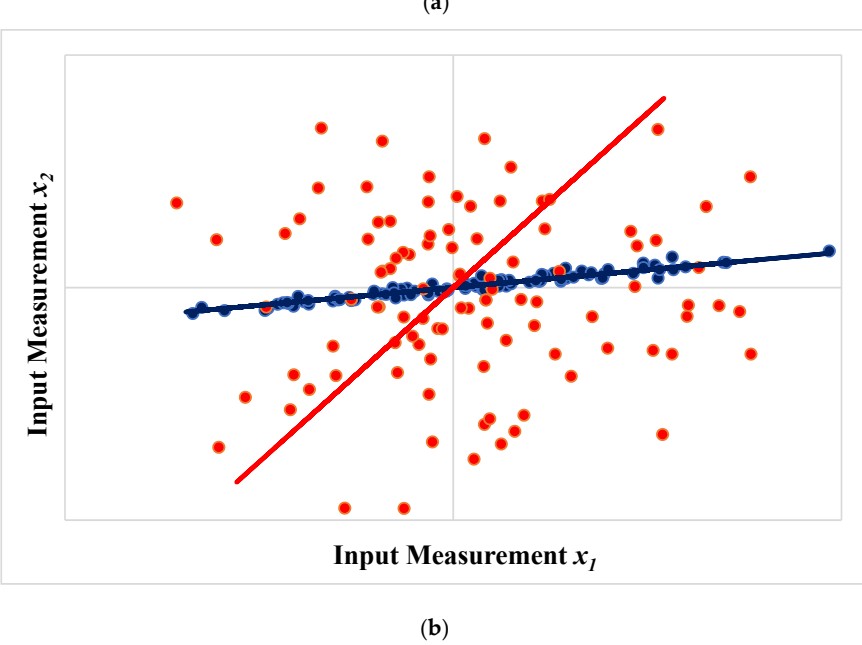

(**b**)

**Figure 1.** Effect of measurement fluctuations on the principal directions. (**a**) Datasets with different input populations. (**b**) Most influential principal directions calculated with standard PCA analysis (red) and the proposed H-PCR procedure (blue).

### 3.2. Analysis of NIR Spectra

As reported previously [18], error and correlation analyses of actual NIR spectral data provided the following results:

- The covariance matrixes of measurement fluctuations depend on the measurement condition and on the considered spectral region;
- The increase of stirring velocity increases the variability of spectral measurements because of the unavoidable shaking of mechanical parts and possible formation of air bubbles (as in real monitoring environments);
- The increase of temperature increases the variability of spectral measurements, possibly because of the lower system viscosity and increased rates of air bubbles formation;
- Spectral measurements in the NIR region are subject to strongly correlated fluctuations, so that measurement error fluctuations are not independent, which must be considered during quantitative analyses.

Therefore, based on the previous paragraphs, the use of H-PCR techniques for model building and calibration can be necessary, as obtained spectral measurements in the NIR region were subject to non-uniform and collinear measurement uncertainties that depended on measurement conditions. Similar characteristics can also be observed in other experimental systems, as already discussed. It is important to emphasize that the qualitative behaviors of first-derivative and second-derivative spectra of the analyzed spectral data were very similar to the behavior of the crude spectra, being sensitive to changes of temperature and stirring speeds and presenting variances and covariances that responded to changes of wavelengths, temperature and stirring speeds. Monteiro et al. (2017) [18] showed that the use of standard pretreatment techniques did not change the overall behavior of the spectral variability and it could be noted that differences of at least one order of magnitude could be observed for computed variances in all cases.

### 3.2.1. Classical Linear Squares (CLS)

A simple linear calibration was performed with the selected data with the help of the software *Statistica 6.0* [50]. The used model was defined as

$$C_x = a + b\,A \tag{30}$$

where $C_x$ is the xylene concentration (vol%), A is the intensity of the absorbance of the first derivative of measured NIR spectra and *a* and *b* are parameters of the calibration model, respectively linear and angular coefficients. Calibration models were initially built with different datasets, using all available data for specific wavelength values. Then, data were grouped in terms of temperature, stirring speed or both for specific wavelength values. Stirring speed and temperature variations significantly affected the performances of the calibration models, because of the modification of the statistical behavior of the collected spectral data. For this reason, most models were also built for constant temperature and stirring speed conditions. The obtained results showed that the model performances were usually bad and significantly influenced by the wavelength, temperature and stirring variations. In order to illustrate this behavior, Figure 2 shows the model performance for stirring speed of 350 rpm at the wavelength of 2052 nm and Figure 3 shows the model performance for temperature of 60 °C at the wavelength of 2052 nm. Calculations have been made for different wavelengths and all of them presented similar behaviors at the distinct operation conditions, showing that improved modeling procedures should be pursued.

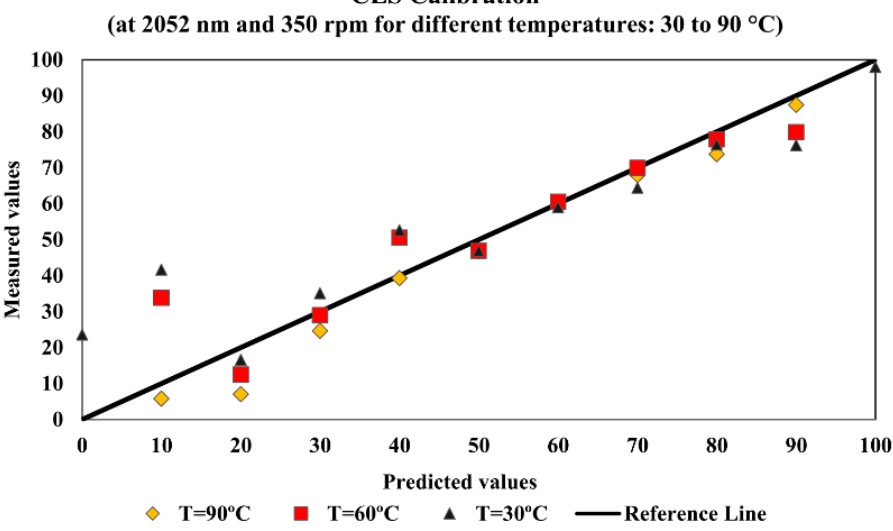

**Figure 2.** Simple linear calibration model performance at 2052 nm for stirring speed of 350 rpm.

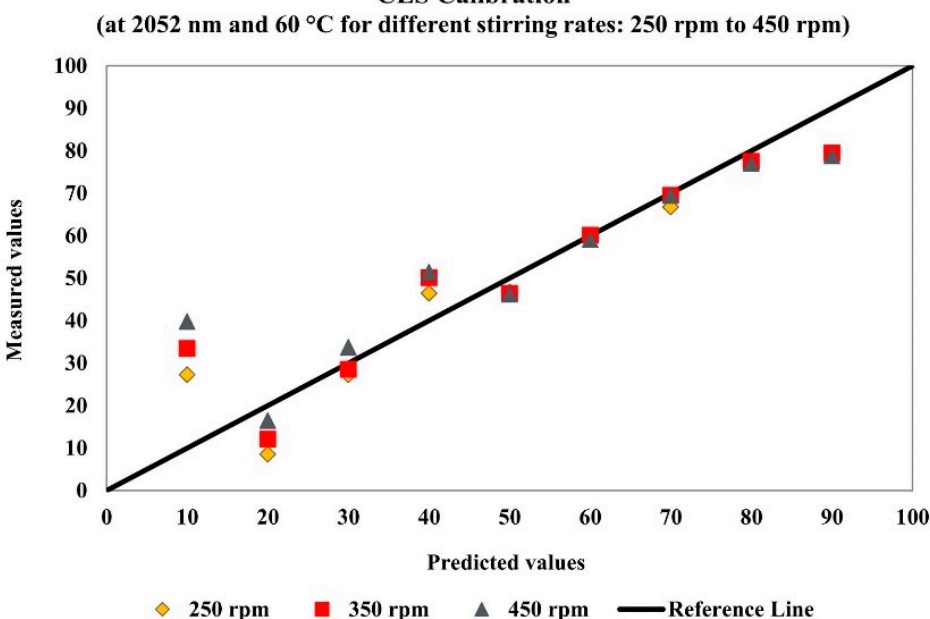

**Figure 3.** Simple linear calibration model performance at 2052 nm for temperature of 60 °C.

### 3.2.2. Analysis of the $V_x$ Matrix

In order to analyze the structure of the covariance matrix of measurement error, $V_x$, 80 spectral measurements were performed for a solution containing 20 vol% xylene and 80 vol% toluene at 30 °C and stirring velocity of 250 rpm. Then, the 79 eigenvectors and eigenvalues of the $V_x$ matrix were calculated. From this analysis it could be concluded that not more than four eigenvalues were sufficient to explain 99% of the total variability of the problem, clearly indicating the small number of sources of variability in the system (when compared to the number of spectral responses, *m*) and the numerical problems related to the inversion of $V_x$, as described previously and completely neglected in the available NIR literature. This can be regarded as a particularly important result, because it clearly indicates that suitable pseudo-inverses should be obtained to represent the inverse of $V_x$ in maximum likelihood calibration procedures.

The four eigenvectors that concentrated 99% of the problem variability are presented in Figure 4. This result can be regarded as very important because it clearly indicates that the usual principal directions, as calculated with standard PCR techniques, may not constitute the most suitable set of directions for modeling purposes in heteroscedastic problems, given the effect of varying measurement errors on the final performances of calibration models. The accumulated variability for each eigenvector included in the analysis is presented in Figure 5.

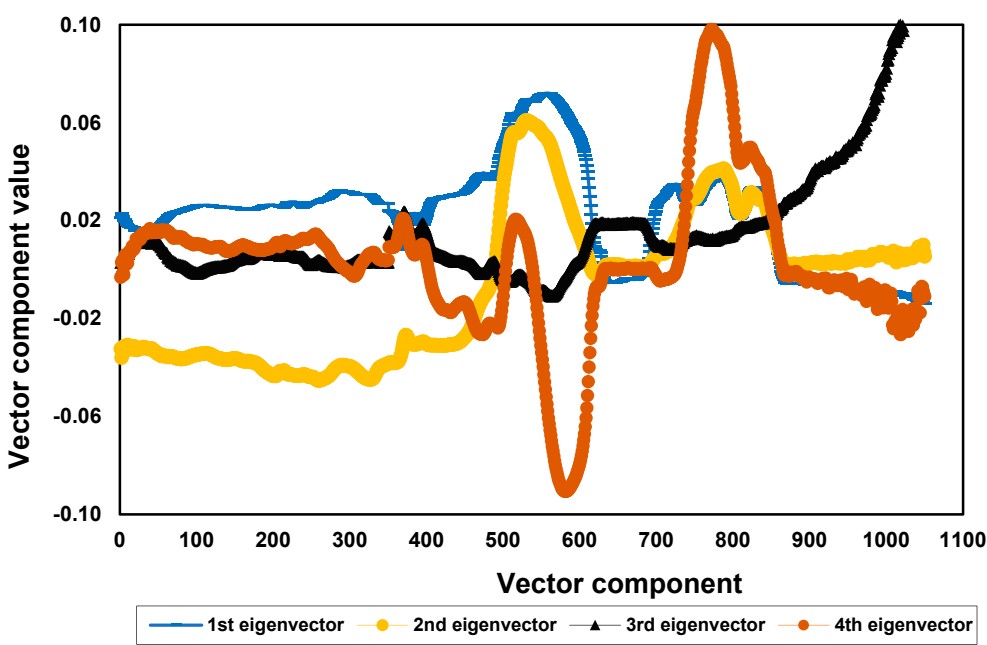

**Figure 4.** The four eigenvectors of the covariance matrix of NIR spectral measurement fluctuations evaluated for solutions containing 20 vol% xylene and 80 vol% toluene at 30 °C and stirring velocity of 250 rpm.

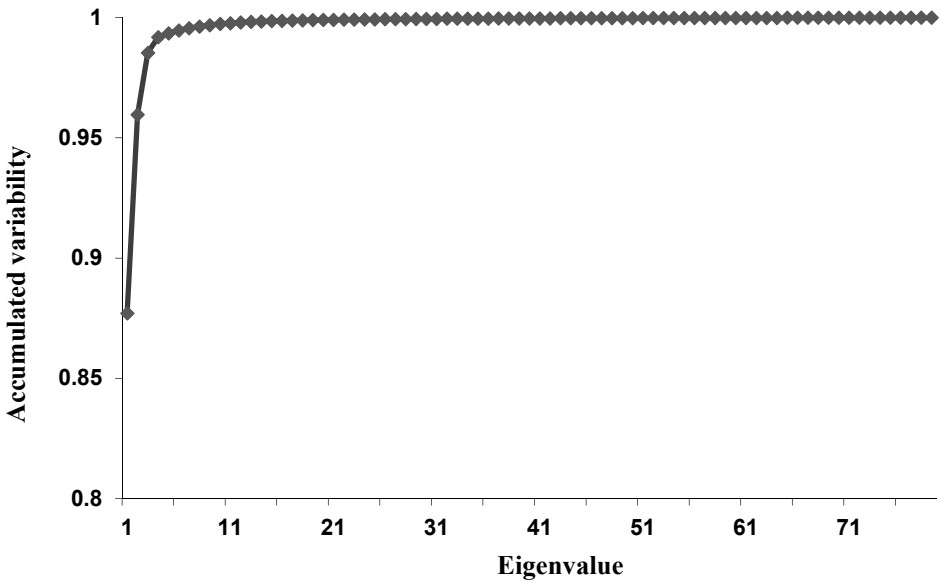

**Figure 5.** Total accumulated variability for xylene–toluene mixture analysis.

### 3.3. PLS, PCR and H-PCR Calibrations

The principal directions were calculated using PLS, PCR and H-PCR methods. In the cases of PLS and H-PCR calibrations, the principal directions were calculated with help of the stochastic numerical procedure developed in Section 2.2, considering the intrinsic heteroscedastic nature of experimental variances. It could be noted that the directions calculated when **Vx** variations were taken into consideration were very different from the directions calculated when these variations were neglected. Therefore, information regarding the variations of **Vx** can definitely affect the final calibration model performances, as shown by Monteiro et al. (2017) [18] for the first time and illustrated in Section 3.1. As observed experimentally, differences increased when increasing amounts of information were extracted (in other words, differences are more significant for the fifth principal

direction than for the first principal direction). Figure 6a–e presents the five first principal directions, as calculated for calibrations performed for illustrative purposes with PLS, PCR and H-PCR at 90 °C and 450 rpm, showing that variations of **Vx** change the principal directions and can affect the model calibration, clearly indicating that variations of **Vx** should not be disregarded a priori, as usually done and as illustrated in Section 3.1. The less significant modification of the first principal directions is probably related to the fact that the first directions tend to reflect more clearly the absolute variations of the variable coordinates (or variations of average responses), while the other directions tend to reflect more clearly the measurement variability around the main averages, which are more sensitive to the **Vx** values.

The evolutions of the model performances as functions of the number of latent variables, as calculated for calibrations performed for illustrative purposes with PLS, PCR and H-PCR at 90 °C and 450 rpm, are presented in Figures 7–9. As one can see in Figures 7 and 8, the calibration objective function values decreased and the model correlation coefficients increased when the number of latent values increased, as one might already expect. However, the speeds of variation of both objective function values and model correlations with the number of latent values depended on the particularly analyzed numerical technique. When 10% of the original dataset was selected at random and used for cross-validation, 10 latent variables were selected for all the models.

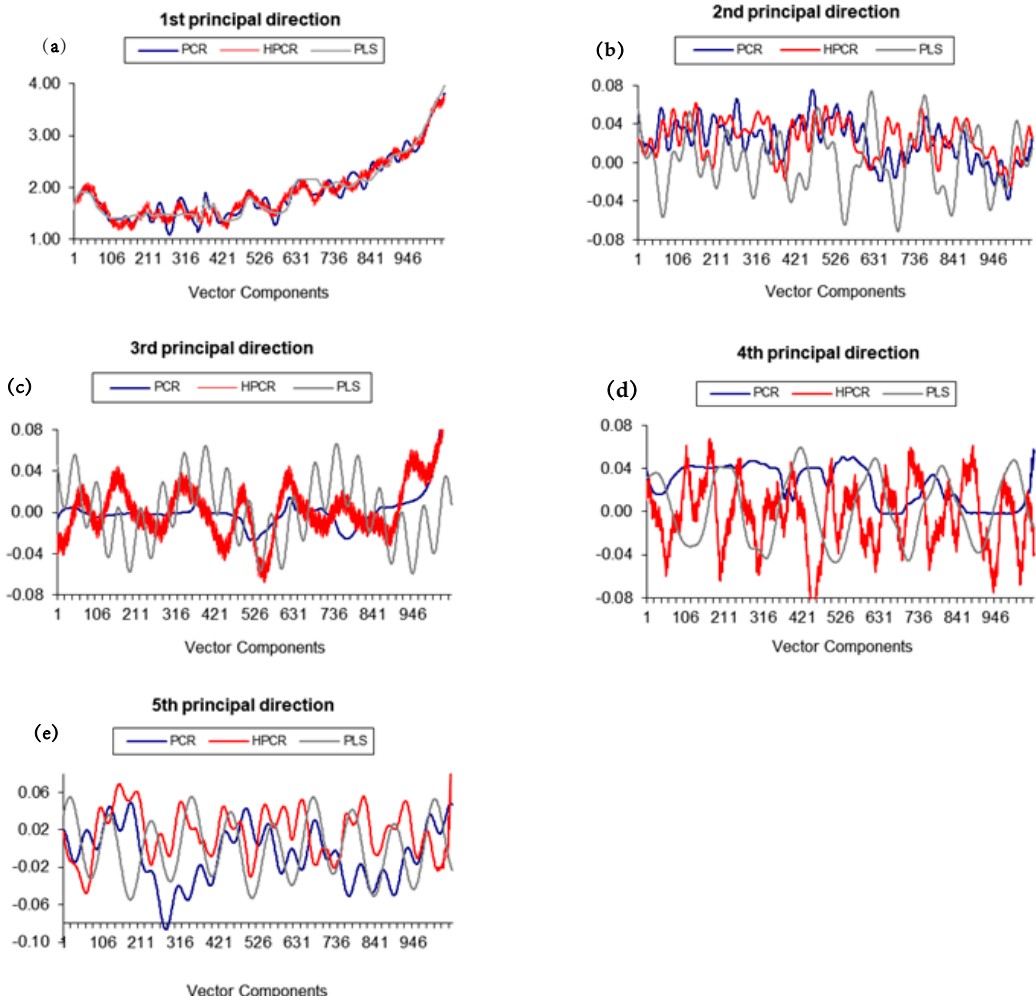

**Figure 6.** Principal directions of model calibrations using PLS, PCR and H-PCR at 90 °C and 450 rpm (**a**) first principal direction; (**b**) second principal direction; (**c**) third principal direction; (**d**) fourth principal direction; (**e**) fifth principal direction.

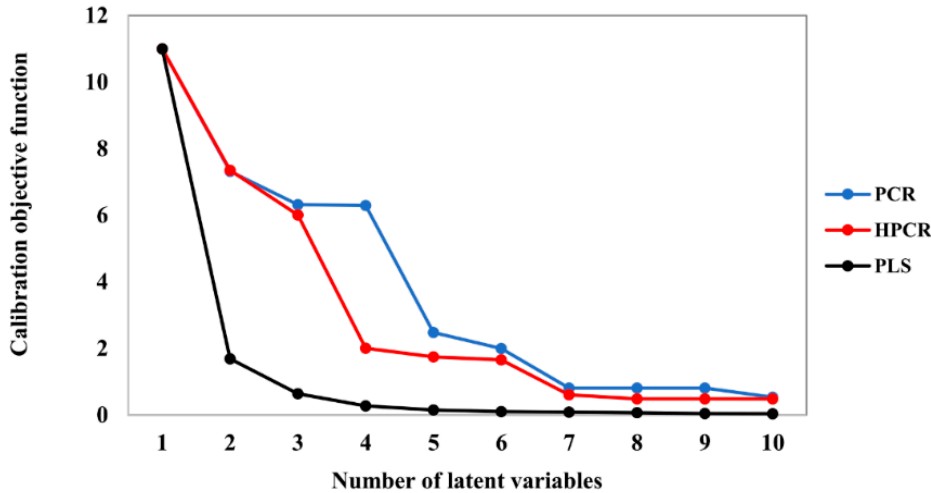

**Figure 7.** Calibration objective functions for PLS, PCR and HPCR models at 90 °C and 450 rpm.

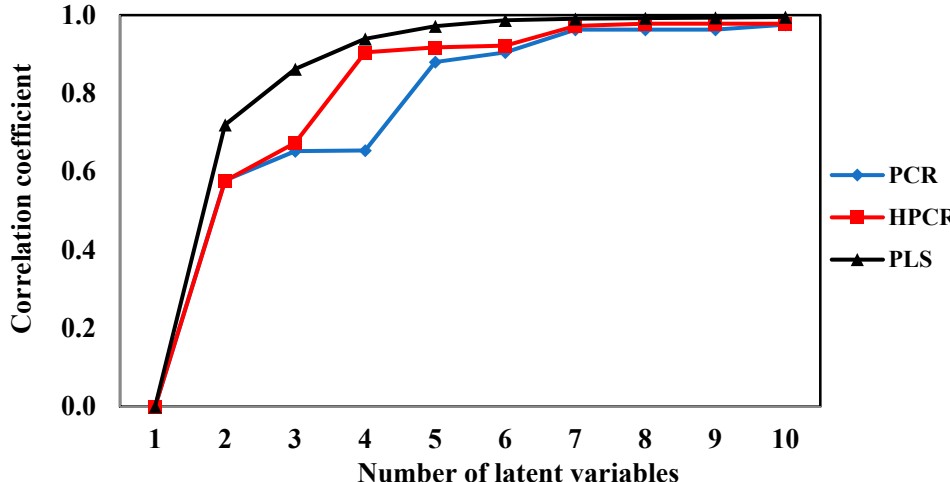

**Figure 8.** Correlation coefficients for PLS, PCR and HPCR models at 90 °C and 450 rpm.

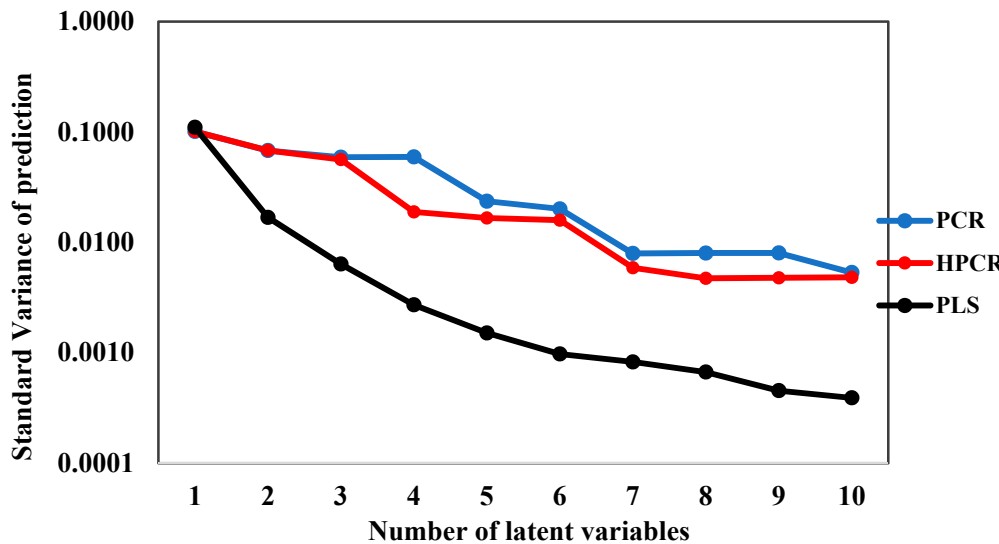

**Figure 9.** Standard variances of prediction for PLS, PCR and HPCR models at 90 °C and 450 rpm.

As one might already expect, obtained correlation coefficients were higher for the PLS approach (Figure 8). This is not surprising because the PLS approach attempts to select the latent variables that maximize the correlation among measured and calculated responses, while the PCR and H-PCR procedures define the latent variables based on some specific features of the input variable space, as already described. Consequently, the PLS technique establishes a sort of threshold of maximum correlation performance for the calibration system based on latent variables. In spite of that, one must observe that calibration performances obtained with the proposed H-PCR approach were always better than the ones obtained with the usual PCR approach and close to the performances of the PLS model when the number of latent variables was higher than three. This clearly shows that the explicit consideration of heteroscedasticity in the numerical formulation can benefit the quantitative calibration step. Table 2 and Figure 10 present the prediction variances of model outputs at the different concentrations used for model calibration at 90 °C and 450 rpm, clearly indicating the heteroscedastic behavior of the prediction variances of model outputs and justifying the use of the heteroscedastic PLS procedure for modeling purposes.

**Table 2.** Data for error prediction for different xylene concentrations at 90 °C and 450 rpm using H-PCR method.

| Concentration [wt Fraction] | Prediction Variance [wt Fraction]$^2$ | F Value * |
|:---:|:---:|:---:|
| 0 | 0.001424 | **4.387** |
| 0.10 | 0.001723 | **5.331** |
| 0.20 | 0.002378 | **7.329** |
| 0.30 | 0.002293 | **7.068** |
| 0.40 | 0.007016 | **21.625** |
| 0.50 | 0.001521 | **4.688** |
| 0.60 | 0.02823 | **8.700** |
| 0.70 | 0.002033 | **6.267** |
| 0.80 | 0.001014 | 3.127 |
| 0.90 | 0.0000418 | 1.289 |
| 1.00 | 0.000324 | 1.000 |

* Limiting F-value for confidence level of 95% is equal to 4.02. Numbers in bold are statistically different from prediction variance measured for c = 1 with 95% of confidence.

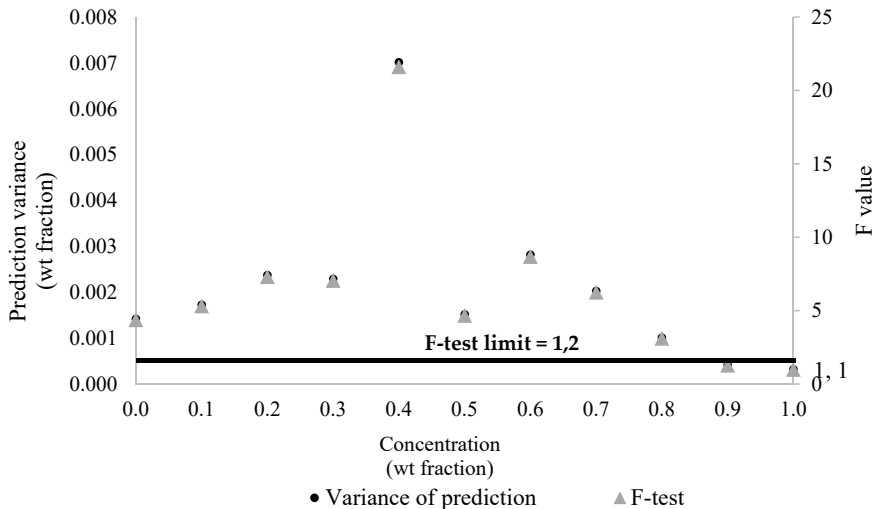

**Figure 10.** Error performance for H-PCR at 90 °C and 450 rpm for different concentrations.

It is important to observe that the use of a high number of latent variables should not be regarded as a disadvantage of the proposed analyses. Firstly, one must observe that the use of 10 latent variables is not bad for practical purposes, as the model is still sufficiently compact for fast and reliable use in real problems in this case. Besides, model performances could be regarded as good in all cases where the number of latent variables was higher than six. Secondly, one must also observe that the selected PLS model provided a correlation of 0.995 for 10 latent variables, while the selected PCR model provided a correlation of 0.975 and the selected H-PCR model provided a correlation of 0.978, which could be regarded as excellent in all cases.

It is also important to observe that a stochastic procedure was used for computation of latent variables and directions in the proposed H-PCR approach. For this reason, it must be noted that convergence of the proposed method was assured, as shown in Figure 11, which shows the first principal directions obtained with the H-PCR model for different numerical runs. It can be noted that the calculated directions (and model responses) were essentially the same in all cases, characterizing the appropriate convergence of the proposed approach.

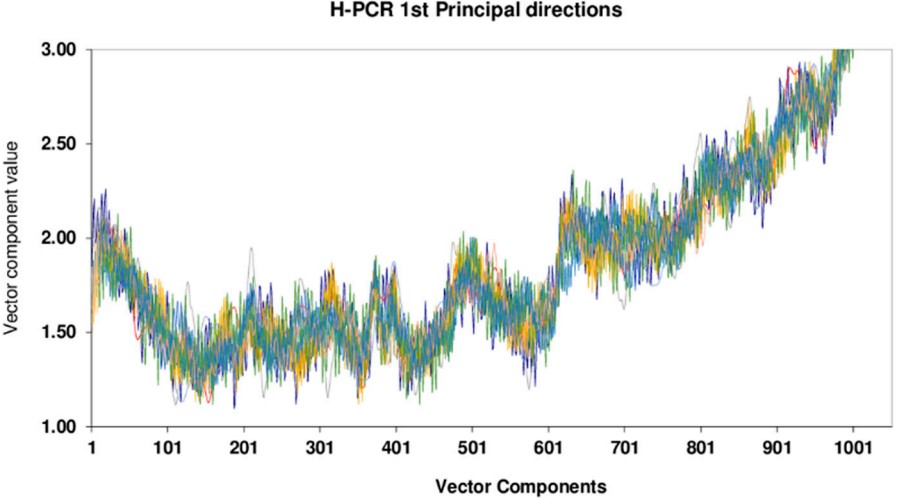

**Figure 11.** The first principal directions for various numbers of latent variables using the H-PCR model for method convergence test.

## 4. Conclusions

In the present work, a new heteroscedastic principal component regression (H-PCR) procedure was presented. By incorporating the variations of the covariance matrix of measurement fluctuations ($\mathbf{Vx}$) into the analysis, the proposed procedure can provide more reliable principal directions and model calibrations. Particularly, a stochastic algorithm was proposed and implemented to allow for H-PCR calculations.

Calibration models were built in a simple experimental problem (mixtures of xylene and toluene) in order to compare the performances of CLS (classical least squares), PCR, H-PCR and PLS (partial least squares) techniques. It was initially observed that CLS calibration models were usually inefficient and unable to deal with the variations of the analyzed experimental problem. Then, it was shown that the principal directions calculated when variations of $\mathbf{Vx}$ are taken into consideration can be quite different from the ones calculated when variations of $\mathbf{Vx}$ are disregarded, indicating the influence of the measurement fluctuations on the model calibration step. Finally, it was shown that H-PCR models can allow for appropriate calibration of NIR data and that identification of the most compact and reliable calibration model can be performed better with the H-PCR technique than with the PCR technique. It is also important to observe that the analysis of the $\mathbf{Vx}$, as obtained experimentally, indicated that few eigenvalues can concentrate most of the system variability, confirming that $\mathbf{Vx}$ can be nearly singular, demanding the use of pseudo-inverse procedures for proper implementation of H-PCR techniques.

**Author Contributions:** Conceptualization, J.C.P., T.d.S.F. and A.d.R.D.M.; methodology, J.C.P., T.d.S.F. and A.d.R.D.M.; software, J.C.P., A.d.R.D.M. and T.d.S.F.; validation, J.C.P., T.d.S.F. and A.d.R.D.M.; formal analysis, J.C.P.; investigation, A.d.R.D.M. and T.d.S.F.; resources, J.C.P., A.d.R.D.M. and T.d.S.F.; data curation, J.C.P., A.d.R.D.M. and T.d.S.F.; writing—original draft preparation, A.d.R.D.M.; writing—review and editing, J.C.P., A.d.R.D.M. and T.d.S.F.; visualization, J.C.P. and A.d.R.D.M.; supervision, J.C.P.; project administration, J.C.P. All authors have read and agreed to the published version of the manuscript.

**Funding:** This research received no external funding.

**Data Availability Statement:** The full set of collected spectral data can be provided for further analyses if required.

**Acknowledgments:** The authors thank CNPq (Conselho Nacional de Desenvolvimento Científico e Tecnológico). CAPES (Coordenação de Aperfeiçoamento de Pessoal de Nível Superior) and FAPERJ (Fundação Carlos Chagas Filho de Apoio à Pesquisa do Estado do Rio de Janeiro) for financial support and scholarships.

**Conflicts of Interest:** The authors declare no conflict of interest.

## Nomenclature

| Symbol | Description |
| --- | --- |
| $y$ | Vector model responses |
| $x_i$ | Available data (or inputs) |
| $n$ | Total number of data |
| $\sigma_{ij}$ | Standard deviation at experimental condition $i$ |
| $\sigma^2$ | Variance |
| $\mu$ | Contracting factor to convergence control |
| $\rho$ | Correlation factor |
| $F_{obj}$ | Objective function |
| $\lambda$ | Eigenvalues |
| $\Lambda$ | Diagonal matrix with eigenvalues |
| $D$ | Matrix with eigenvectors |
| $\varphi i$ | Variability fraction along the $i$th direction |
| $\mathbf{V}_{xi}$ | Matrix of variance |
| $p$ | Principal direction |
| $a_{min}, a_{max}$ | Amplitude interval |
| $b_{min}, b_{max}$ | Lag interval |
| $f_{min}, f_{max}$ | Frequency interval |
| $\delta$ | Tolerance |
| $\sigma_r$ | Reduction factor |

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
