# Peer review of "A Numerical Procedure for Multivariate Calibration Using Heteroscedastic Principal Components Regression"

_processes, doi:10.3390/pr9091686_

Round 1
Reviewer 1 Report
1. This manuscript describes H-PCR techniques for multivariate calibration. The NIR spectrum of mixtures of xylene and toluene with different temperatures and stirring velocities are the experimental target. A serial comparison analysis of PCR, HPCR, and PLS methods are proposed in this article. To my knowledge, the method proposed in this manuscript is novel and has application potential in the chemometric field.
2. Below equation 24, the authors mentioned the parameter δ. Could the authors explain why it was made equal to 1.0x10-4, and the effect of this parameter on the convergence time and accuracy of this model? This could be the recommendation for the reader to determine the setting of the parameter.
3. Chapter 4 seems missing.
4. In figure 4, that needs the labels for the x and y axes. Otherwise, the readers are hard to understand the meaning of this figure.
5. The explanation for figure 6 is probably not clear enough for the readers. Please provide more explanation for those figures, and also enhance the clarity for it, e.g. the reviewer can only see two curves in the figure6. (e).
6. The formats of the reference are inconsistent and inappropriate, e.g. in reference [18] the symbol " is missing.
Reviewer 2 Report
This paper presents an interesting topic, and proposes a new HPCR method. Overall the method is well described and the paper novelty clearly stated.
There are a lot of variables in the paper, although they are all defined in the text, it would be nice to have them in a nomenclature section.
The paper is well-written and easy to read. There are several sentences where the comas appear as dots, particularly in the end of the paper. This must be corrected.
There is a small mistake in the description of the mass storage of the workstation (8 GB is surely wrong).
Overall, I recommend the publication of this paper.
Round 2
Reviewer 1 Report
The manuscript has been sufficiently revised. The quality of the paper has improved after the revision, and I recommend it for publication.
However, in my opinion, there are still some issues that can be improved.
(1) To improve the quality of the pictures. Make the decoration of the figures in the same style. For example, some of the figures are with the frames, and some of them are without.
(2) The upper frame of figure 1 seems missing.
Reviewer 2 Report
I recommend the revised paper for publication in its present form.
Round 3
Reviewer 1 Report
The manuscript has been sufficiently revised. The quality of the paper has improved after the revision, and I recommend it for publication.